# The prevalence of Dupuytren's disease in patients with diabetes mellitus

Sandhya Ganesan[1], Ryan P. Tantone[1], David E. Komatsu [1] & Lawrence C. Hurst [1✉]

## Abstract

**Background** Dupuytren's disease (DD) is a fibroproliferative hand disorder associated with various medical conditions, including diabetes mellitus (DM). The reported prevalence of DM among DD patients varies widely, primarily due to small sample sizes in previous studies.

**Methods** This was a retrospective cohort study using data from the TriNetX Research Database. We analyzed the overall prevalence of DD between 2010 and 2020, comparing the DM, type 1 diabetes mellitus (T1DM), and type 2 diabetes mellitus (T2DM) cohorts. Within the DM group, patients were further categorized based on hemoglobin A1c (HbA1c) values and prescribed anti-diabetic agents (insulin or metformin). We compared the prevalence of DD diagnosis in each group using prevalence ratios and differences.

**Results** There is a higher prevalence of DD in patients with T2DM than in patients with T1DM (relative risk [RR]: 1.641; 95% confidence interval [CI]: [1.356, 1.986]). Among patients with diabetes, there is a higher prevalence of DD in those taking insulin compared to those taking metformin (RR: 0.801, 95% CI: [0.774, 0.83]). The prevalence of DD varies depending on HbA1c levels, with a prevalence of 0.463% in patients having levels within the diabetic range, while lower prevalences of 0.392% and 0.416% are found in patients with prediabetes or uncontrolled diabetes, respectively.

**Conclusions** This study provides further insight into the relationship between DM and DD. These findings may be attributed to the increased accumulation of advanced glycosylated end products (AGEs) in patients with diabetes. Future research exploring the connection between AGE accumulation and DD development may enhance our understanding of the relationship between DD and DM.

## Plain language summary

Dupuytren's disease (DD), commonly known as Dupuytren's contracture, is a disorder of the hand that has been associated with various conditions including diabetes. The relationship between the two has not been studied in large populations; therefore, we used a large electronic medical record database to better understand the association between these two conditions. Our analyses show that within the population of patients with diabetes, DD is more common in patients with adult-onset diabetes and patients with blood sugar levels corresponding to moderate diabetes. This finding may be related to biochemical changes in the body as a result of elevated blood sugar levels found in these patients. Future investigation into this biochemical change may contribute further to our understanding of the relationship between these two conditions.

[1] Department of Orthopedics, State University of New York at Stony Brook, HSC T18 Room 092, Stony Brook, NY 11794-8181, USA.
✉email: Lawrence.Hurst@stonybrookmedicine.edu

Dupuytren's disease (DD) is a benign fibroproliferative disorder of the hand first described by Felix Platter in 1614[1]. The disease was later characterized by Baron Guillaume Dupuytren, who performed the first documented open fasciotomy on June 12, 1831[2]. Since then, much has been done to investigate the pathogenesis, treatment, and potential cure for this disease. While the exact cause of DD remains unclear, multiple genetic and environmental factors have been suggested to play a role[3,4]. Additionally, DD has been reported to occur at higher rates in patients that have other diseases such as diabetes mellitus (DM), alcoholism, and epilepsy[4].

Of these diseases, DM is considered an important risk factor. Just fifty-two years after Dupuytren's first fasciotomy, two French writers, Cayla and Viger, noted a connection between DD and DM[5,6]. Despite this early report, there are still no large cohort studies reporting on the association between DD and DM. We reviewed the literature and identified 26 studies with a total of 10,427 patients with DD and/or DM[7–32]. Moreover, 3941 of these patients from our literature search were duplicates between a systematic review and the original studies from which they came[7]. Based on these studies, the prevalence of these two diseases varies widely from 2 to 63%[7]. The wide variation in the prevalence data is affected by the different populations studied and the varied definitions and methods of diagnosing DD and DM.

In this study, data from a TriNetX cohort of 88,661,090 patients containing 100,218 patients with DD and 6,019,023 patients with DM are evaluated with the goal of eliciting a more accurate association between Dupuytren's disease and diabetes. The prevalence of DD is 0.11%, confirming the condition's orphan disease status. Within the 100,218 Dupuytren's patients in this study, the prevalence of DM is 14%. Analysis of the prevalence of DD within different cohorts of patients shows a higher prevalence of DD in patients with T2DM than in patients with T1DM. This is also reflected in the different medications predominantly taken by patients with either type 1 or type 2 diabetes, as there is a higher prevalence of DD in patients with diabetes taking metformin compared to patients with diabetes taking insulin. Finally, patients with diabetes with HbA1c levels falling within a diabetic range have the highest prevalence of DD when compared to patients with HbA1c levels in a prediabetic or uncontrolled diabetes range.

## Methods
**Dataset.** This was a retrospective cohort study of patients with DD and DM. The data were collected from the TriNetX Research Network, which provides access to electronic medical records from approximately 100 million patients from 65 healthcare organizations. All data displayed on the TriNetX Platform in aggregate form, or any patient-level data provided in a dataset generated by the TriNetX Platform, only contains de-identified data. Because this study used only de-identified patient records and did not involve the collection, use, or transmittal of individually identifiable data, this study was exempted from approval by the Stony Brook University Institutional Review Board in accordance with 45 CFR § 164.514. As this study was exempt, the requirement for informed consent was waived. Further information regarding TriNetX's data can be found at the following website: https://support.trinetx.com/hc/en-us/sections/360000928753-About-the-Data.

The current study included EMR data from patients included in the TriNetX Research Network as of July 17, 2022. Only aggregate EMR was used in this study, as access to individual patient records was not possible using the TriNetX database. The DD cohort included patients with a diagnosis code for palmar fascial fibromatosis (ICD-10 code: M72.0). The DM cohort included those with a diagnosis code of either type 1 diabetes mellitus (T1DM) or type 2 diabetes mellitus (T2DM) (ICD-10 codes: E10, E11). Patients with diabetes mellitus due to an underlying condition, drug or chemical-induced diabetes mellitus, or other specified diabetes mellitus (ICD-10 codes: E08, E09, E13) were excluded from this cohort. Two additional cohorts were created to separate patients with T1DM and patients with T2DM. The T1DM cohort included patients with ICD-10 diagnosis code E10 and excluded patients with ICD-10 diagnosis codes E08, E09, E11, and E13. The T2DM cohort included patients with ICD-10 diagnosis code E11 and excluded patients with ICD-10 diagnosis codes E08, E09, E10, and E13. ICD-10 code E12, malnutrition-related diabetes mellitus, was not reported on the TriNetX network and therefore excluded from all cohorts.

**Statistical analysis.** Demographic information, including age, gender, and race, was collected and compared across cohorts. Differences in demographic characteristics between cohorts were compared to determine significant differences. Mean age was compared using two-tailed Student's $t$-tests, while gender and race distributions were compared using Chi-squared tests. The prevalence of DD and cumulative cases were analyzed during the years 2010–2020 and compared across DM, T1DM, and T2DM cohorts. The overall difference in the prevalence of DD was also analyzed across DM, T1DM, and T2DM cohorts. Patients were further stratified into three groups according to HbA1c values: prediabetes (HbA1c < 6.5%), diabetes (6.5% < HbA1c < 7.5%), or uncontrolled diabetes (HbA1c > 7.5%). Further stratification was performed based on prescribed anti-diabetic agents: insulin or metformin. The prevalence of DD was calculated as (patients with DD)/(patients in the cohort). The prevalence of DD associated with each group was assessed and compared to one another to determine additional factors associated with Dupuytren's disease using prevalence differences and prevalence ratios. Statistical analyses were conducted on the TriNetX platform, which utilizes a combination of JAVA™, R,21 and Python™ programming languages in addition to Microsoft Excel (Version 16.66.1). Patient records more than 20 years old were automatically excluded from analysis by the TriNetX platform.

**Reporting summary.** Further information on research design is available in the Nature Portfolio Reporting Summary linked to this article.

## Results
The overall population included in TriNetX consists of 88,661,090 patients. Demographic data for each cohort are shown in Table 1. The prevalence of these conditions in the overall TriNetX population is as follows: 0.11% for DD, 6.8% for DM, 0.20% for T1DM, and 6.2% for T2DM. The total number of patients in the DM cohort is larger than the sum of the patients in the T1DM and T2DM cohorts because certain patients had a general diabetes mellitus diagnostic code in their chart without specification of which type.

The distribution of age and gender for patients within the DD cohort is shown in Fig. 1. The distribution of age and gender, as previously reported by Mikkelson in the literature, mirrors the distribution of 100,218 patients included in Fig. 1[33]. Additionally, there are significantly more men than women within the DD cohort as compared to the overall TriNetX patient population ($p < 0.01$).

Comparisons were then made between cohorts to identify significant differences in demographic characteristics. When comparing the DD and DM cohorts, there is a significant

**Table 1 Demographic characteristics for Dupuytren's disease (DD), diabetes mellitus (DM), type 1 DM (T1DM), and type 2 DM (T2DM) cohorts.**

|  | DD | DM | T1DM | T2DM |
|---|---|---|---|---|
| Total patients | 100,218 | 6,019,023 | 178,168 | 5,537,511 |
| Mean age (years) | 70 ± 12 | 64 ± 18 | 43 ± 23 | 65 ± 17 |
| Gender |  |  |  |  |
| Male | 60,130 (60%) | 2,949,321 (49%) | 90,865 (51%) | 2,713,380 (49%) |
| Female | 40,087 (40%) | 3,069,701 (51%) | 87,302 (49%) | 2,824,130 (51%) |
| Race |  |  |  |  |
| White | 85,185 (85%) | 3,671,604 (61%) | 112,245 (63%) | 3,377,881 (61%) |
| Black | 3006 (3%) | 1,083,424 (18%) | 19,598 (11%) | 996,751 (18%) |
| Asian | 1002 (1%) | 180,570 (3%) | 1781 (1%) | 166,125 (3%) |
| Unknown | 11,023 (11%) | 1,083,424 (18%) | 44,542 (25%) | 941,376 (17%) |

The total number of patients, including the mean age, gender distribution, and race distribution, were reported for each cohort.

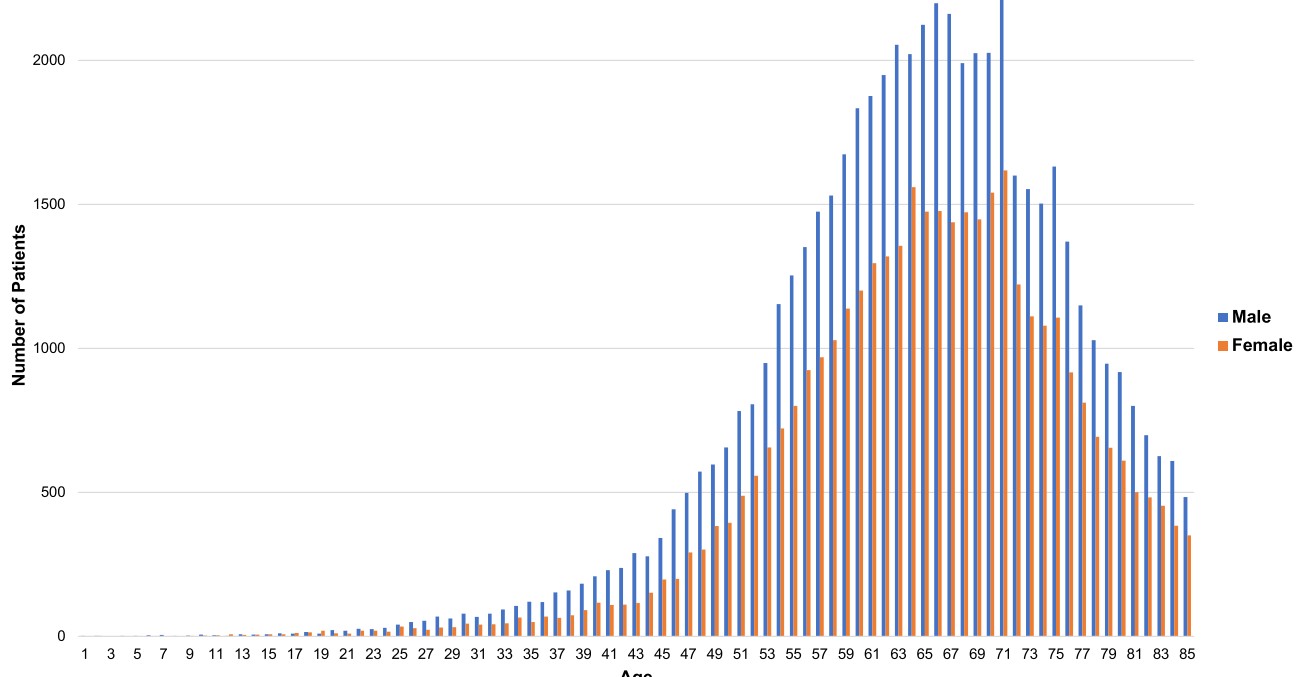

**Fig. 1 Age distribution of patients with Dupuytren's disease (DD) separated by gender.** A total of 92,838 patients are shown on this graph, with 7380 patients aged 86 and older not plotted. This distribution mirrors previous reports of DD in the literature.

difference in the mean age of patients, with DD patients being on average older than DM patients ($p < 0.01$). A significant difference is also found between the gender distribution of the cohorts, with the DD cohort having a higher percentage of males than the DM cohort ($p < 0.01$). Finally, a significant difference is found in the race distribution of the cohorts ($p < 0.01$). Notably, the DD cohort has a higher percentage of White patients than the DM cohort.

When comparing the DD and T1DM cohorts, there is a significant difference in the mean age of patients, with DD patients being on average older than T1DM patients ($p < 0.01$). A significant difference is also found between the gender distribution of the cohorts, with the DD cohort having a higher percentage of males than the T1DM cohort ($p < 0.01$). Finally, a significant difference is found in the race distribution of the cohorts ($p < 0.01$). Notably, the DD cohort has a higher percentage of White patients than the T1DM cohort.

When comparing the DD and T2DM cohorts, there is a significant difference in the mean age of patients, with DD patients being on average older than T2DM patients ($p < 0.01$). A

significant difference is also found between the gender distribution of the cohorts, with the DD cohort having a higher percentage of males than the T2DM cohort ($p < 0.01$). Finally, a significant difference is found in the race distribution of the cohorts ($p < 0.01$). Notably, the DD cohort has a higher percentage of White patients than the T2DM cohort.

The prevalence of DD within patients with diabetes and the overall patient population within TriNetX was compared by age group, as shown in Fig. 2. For all age groups 10–14 years and above, there is a significantly higher prevalence of DD within patients who have diabetes as compared to the overall population ($p < 0.05$).

The prevalence of DD in the DM, T1DM, and T2DM cohorts was collected from years 2010 to 2020, as shown in Fig. 3. The majority of DD cases are found in the T2DM ($n = 5,537,511$) cohort, likely due to a larger population than the T1DM cohort ($n = 178,168$). The cumulative cases of Dupuytren's diagnoses over the same time period are shown in Supplementary Table 1.

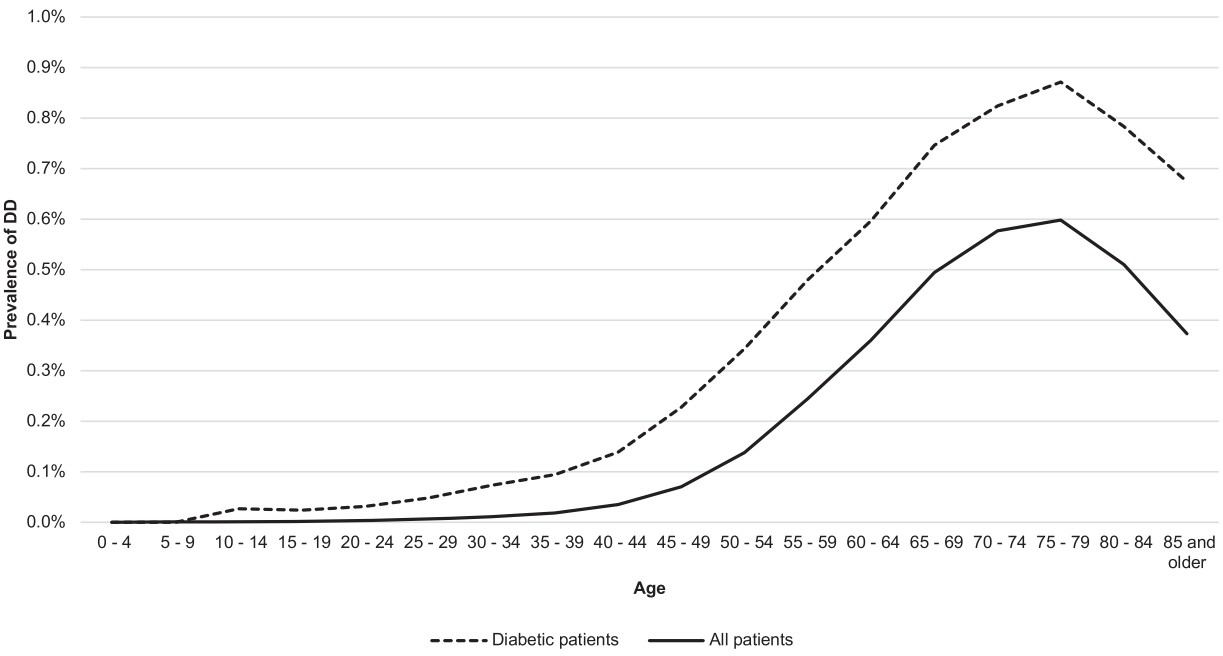

**Fig. 2 Comparison of Dupuytren's disease (DD) prevalence between patients with diabetes and the overall population.** There is a significantly greater prevalence of DD within patients who have diabetes as compared to the overall population. This trend can be seen in age groups 10–14 and above.

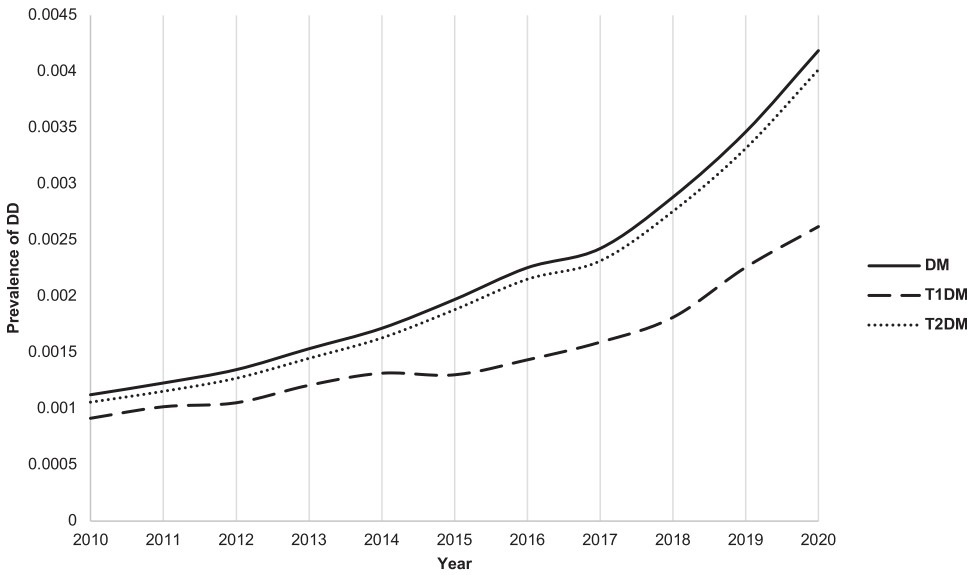

**Fig. 3 Prevalence of Dupuytren's disease (DD) within the diabetes mellitus (DM), type 1 DM (T1DM), and type 2 DM (T2DM) cohorts between the years 2010–2020.** The cases of DD within the DM cohort are largely made up of the cases in the T2DM cohort.

The prevalence of DD was calculated for each cohort and was further analyzed based on HbA1c value and anti-diabetic medication. The prevalence within the DM, T1DM, and T2DM groups are shown in Table 2, with the first column representing the number of patients in the overall DM, T1DM, or T2DM cohort and the second column representing patients within that cohort with a DD diagnosis.

As HbA1c lab values are correlated with the severity of diabetes, we then stratified DM patients by these values and determined their risk of having a DD diagnosis. A smaller subset of patients with DM was included in this analysis because HbA1c values were not available for all patients. Lab values were used to categorize patients into one of three groups: prediabetes (HbA1c < 6.5%), diabetes (6.5% < HbA1c < 7.5%), or uncontrolled diabetes (HbA1c > 7.5%). The prevalence within each HbA1c

**Table 2 Prevalence of Dupuytren's disease (DD) within cohorts of patients with diabetes.**

|  | Patients in cohort | Patients with DD | Prevalence |
|---|---|---|---|
| DM | 5,989,055 | 15,729 | 0.263% |
| T1DM | 171,959 | 271 | 0.150% |
| T2DM | 5,544,341 | 13,779 | 0.249% |

The prevalence of DD was calculated for the diabetes mellitus (DM), type 1 DM (T1DM), and type 2 DM (T2DM) cohorts.

group is shown in Table 3, with the first column representing the number of patients in the overall prediabetes, diabetes, or uncontrolled diabetes cohort and the second column representing patients within that cohort with a DD diagnosis.

**Table 3 Prevalence of Dupuytren's disease (DD) based on hemoglobin A1c (HbA1c) levels.**

| | Patients in cohort | Patients with DD | Prevalence |
|---|---|---|---|
| Prediabetes (HbA1c < 6.5%) | 1,737,379 | 6805 | 0.392% |
| Diabetes (HbA1c 6.5–7.5%) | 1,446,835 | 6695 | 0.463% |
| Uncontrolled diabetes (HbA1c > 7.5%) | 1,424,822 | 5926 | 0.416% |

The prevalence of DD was calculated for different HbA1c values categorized as prediabetes (HbA1c < 6.5%), diabetes (6.5% < HbA1c < 7.5%), or uncontrolled diabetes (HbA1c > 7.5%).

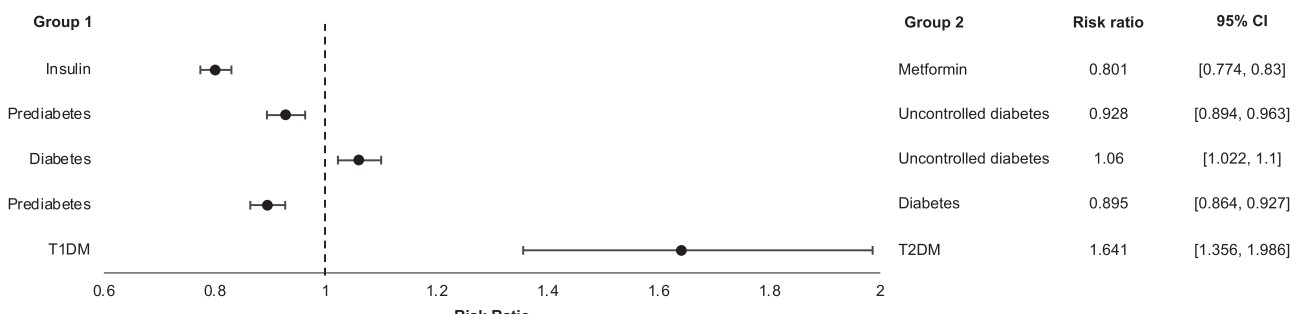

**Fig. 4 Prevalence ratios and 95% confidence intervals (CI) between age and gender-matched cohorts.** Forest plot showing prevalence ratios and 95% confidence intervals for having a Dupuytren's disease (DD) diagnosis after matching for age and gender. Prevalence ratios were calculated as Group 1 vs. Group 2. Prevalence ratios less than one mean Group 1 has a higher prevalence of DD, whereas prevalence ratios greater than one mean Group 2 has a higher prevalence of DD. All prevalence ratios are significant.

Finally, the prevalence of DD in patients taking either insulin or metformin was calculated. The type of anti-diabetic medication that patients take is related to the type of diabetes they are diagnosed with. The T1DM and T2DM cohorts in TriNetX show that a higher percentage of patients with T1DM are prescribed insulin than those with T2DM. A higher percentage of patients with T2DM are prescribed metformin than those with T1DM. These two medications do not account for all anti-diabetic agents being taken, and patients may often take more than one medication to control their diabetes. Out of the total 2,378,452 patients with diabetes taking insulin, 5987 patients have a DD diagnosis, resulting in a prevalence of 0.052%. Out of the total 2,200,152 patients with diabetes taking metformin, 7311 patients have a DD diagnosis, resulting in a prevalence of 0.332%.

In order to make direct comparisons of prevalence between groups, prevalence ratios and prevalence differences were calculated. Groups were matched by age and gender when performing the analysis, meaning fewer patients were used for comparison than in the entire cohort. After matching, a comparison between the prediabetes and diabetes cohorts shows a significant 0.047% higher prevalence for patients within the diabetes cohort ($p < 0.0001$). Comparison between diabetes and uncontrolled diabetes cohort shows a significant 0.026% higher prevalence for patients within the diabetes cohort ($p = 0.0017$). Finally, a comparison between the prediabetes and uncontrolled diabetes cohort shows a significant 0.03% higher prevalence for patients within the uncontrolled diabetes cohort ($p < 0.0001$).

Prevalence ratios between specific cohorts were also calculated after matching for age and gender between cohorts. The forest plot in Fig. 4 summarizes the prevalence ratios and 95% confidence intervals between selected cohorts. Prevalence ratios were calculated by dividing the prevalence of having a DD diagnosis in one group (Group 1 in Fig. 4) by the prevalence of a comparator group (Group 2 in Fig. 4). The dashed vertical line on the plot in Fig. 4 represents a prevalence ratio of 1. Points to the left of this vertical line represent prevalence ratios below 1, indicating that patients within Group 2 of the comparison carry a higher prevalence for a DD diagnosis. Points to the right of this vertical line represent prevalence ratios above 1, indicating that there is a

higher prevalence of DD in Group 2. Patients within Group 2 of the comparison carry a higher risk for a DD diagnosis. All comparisons yielded significant prevalence ratios, as evidenced by the 95% confidence intervals excluding 1.

## Discussion

DD is a complex disorder that, over time, can lead to increasing hand deformity in patients. This, in turn, can lead to a loss of finger motion and hand functionality, creating significant issues with everyday activities. Due to this, there has been a significant investigation into the pathogenesis, risk factors, and potential treatments for the disease.

Previous studies have reported the prevalence of DD to range from 0.6 to 31.6%[1,3,34]. It has been reported to be most prevalent in Caucasians, with previous studies reporting prevalence rates ranging from 75 to 90% for Caucasians, 5 to 8% for African Americans, and 1% or less for Asians[34–36]. The mean age of patients with a diagnosis has been reported to be 50–70 years, with a male-to-female ratio ranging from 1:1 to 2:1, depending on the age of the cohort examined[35,37]. This is consistent with our results. In our study, the mean age of patients with DD was 70, with 60% of patients being male patients and 40% female. In our cohort, 85% of the patients are Caucasian, 3% are African American, and 1% are Asian. Our cohort consists of 100,218 patients with DD, the largest cohort studied to date, and provides further validation of the previously published statistics. Within the overall TriNetX population, the prevalence of DD is found to be 0.11% which is lower than what has been reported in previous studies. These patient records come from various sources ranging from outpatient physician offices to emergency rooms across North America, allowing for the study of a larger, more generalized population than what is possible with patient records from a single hand surgeon or a smaller geographical region. Additionally, given the generalized source of EMR data, it is possible that the incidence found in the present study was lower due to a DD being overlooked or overshadowed by an unrelated complaint for which they presented to the hospital or doctor's office.

Multiple risk factors have been implicated in the incidence and pathogenesis of DD. Commonly cited risk factors include alcoholism, HIV, smoking, seizure disorder, and DM[20]. Of these, the relationship between DM and DD has been extensively investigated. It has been reported that patients with diabetes have a 3.1 times greater chance of developing DD, regardless of the patient's gender[8,38–40]. One study showed an incidence of DD in patients with diabetes to be as high as 20%[3]. In the senior clinical author's case logs for DD, 14% of patients had DM. The association between these two diseases has been theorized to be related to the pathogenesis of each disease. DM leads to the formation of advanced glycation end products (AGEs), which are also associated with other systemic fibroses. It has been shown that patients with diabetes and DD have higher levels of AGEs than controls[8]. The increased AGEs in DD increase fascial stiffness, which through mechanotransduction, accentuates Dupuytren's contractures[9,41]. Additionally, the hyperglycemia associated with chronic diabetes can further contribute to systematic increases in fibrosis[42]. An example of this is diabetic cheiroarthropathy or limited joint mobility syndrome (LJMS), which is seen in 30–40% of patients with chronic diabetes[43,44]. LJMS is diagnosed by the clinical features of progressive pain-free joint stiffness in the hands and feet, impaired grip and fine motor, and a positive prayer sign, as shown in Fig. 5[43]. This hyperglycemia and fibrosis have been further associated with TGFβ activation, increased WNT signaling, and matrix metalloproteinase (MMP) dysregulation. This increased WNT and TGFβ activity, along with MMP dysregulation, has also been strongly implicated in the pathogenesis of DD[8,38,40,41,45–50].

Our study focuses on the incidence, prevalence, and association of DD and DM. As stated previously, the reported prevalence of DD and DM ranges from 14 to 31%[7]. The senior clinical author's case logs found an incidence of 14%. Previous studies have demonstrated a slightly higher, but not statistically significant, association between patients with T1DM and DD compared to those with T2DM[7]. Our study identifies a higher prevalence of DD within the population of patients with T2DM compared to the population of patients with T1DM. Specifically, the prevalence of DD of 0.249% in patients within the T2DM cohort, and the prevalence of DD within the T1DM cohort is 0.0150%. While important, this finding may be confounded by several factors. First, the T1DM cohort has a mean age of 43, the youngest of all the cohorts. As DD occurs most commonly in patients 50 and older, the younger age of the T1DM cohort may underestimate the prevalence of DD in this group. Second, over the past decade, the prevalence of T2DM has increased while the prevalence of T1DM has remained stable[51,52]. This trend is reflected in our results, which shows that the prevalence of DD from 2010 to 2020 increased more in the T2DM cohort compared to the T1DM cohort. Finally, it has been shown that patients with T2DM have a higher level of AGEs and associated comorbidities, two factors associated with DD[53,54]. In the general population, AGEs accumulate with older age. Since patients within the T1DM cohort are on average younger than patients within the T2DM cohort, the lower prevalence of DD found in the T1DM cohort may be related to the lack of time to accumulate AGEs and other comorbidities in general. This association could have further influenced our results.

Our study also demonstrates an increased prevalence of DD in patients depending on the severity of their diabetes, as measured by HbA1c. Of note, the prevalence values found within the three cohorts based on HbA1c values (prediabetes, diabetes, and uncontrolled diabetes) are higher than the prevalence values found within the three overall cohorts of patients with diabetes (DM, T1DM, and T2DM). This is likely due to the smaller subset of patients for which

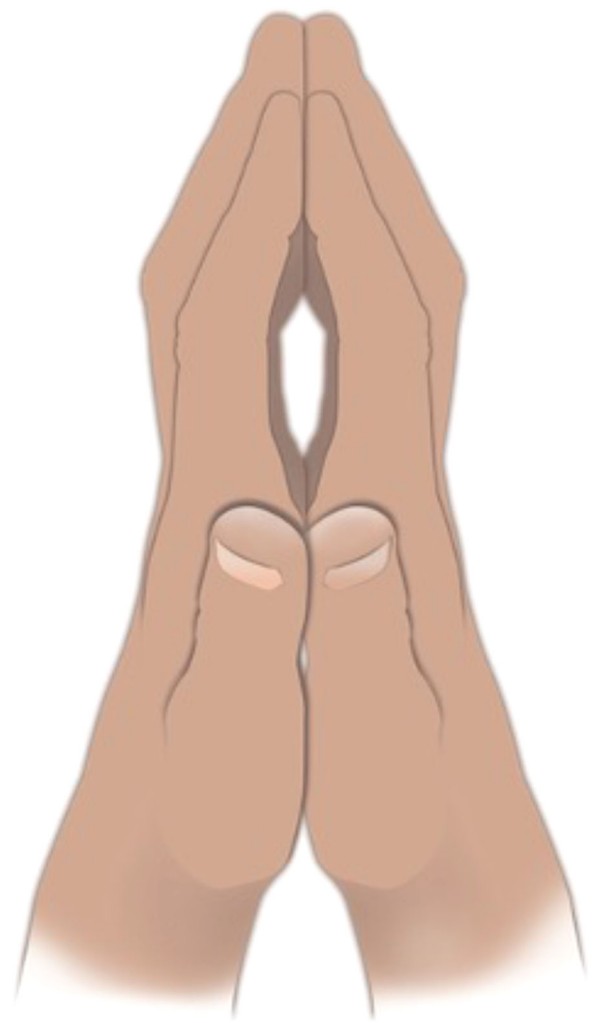

**Fig. 5 Prayer sign.** Positive prayer sign of limited joint mobility syndrome with painless proximal interphalangeal joint contractures not associated with Dupuytren's disease. Positive prayer sign seen in type 1 and 2 diabetes.

HbA1c values were available within TriNetX, as not all patients with diabetes had HbA1c lab values reported in TriNetX. We demonstrate a significant 0.047% increased prevalence of DD in patients with diabetes compared to those with prediabetes. Additionally, there is a significant 0.026% increased prevalence of DD in patients with uncontrolled diabetes compared to those with controlled diabetes. However, this prevalence difference is not as large as the prediabetes and diabetes cohort. In regard to disease pathology, this may be attributed to the amount of AGEs produced in each cohort. It has been shown that AGEs increase with increasing HbA1C levels[55]. Therefore, the difference in AGEs present in those with prediabetes compared to diabetes could be more significant than those with controlled versus uncontrolled diabetes. Future studies should investigate a threshold at which AGE levels have a maximal impact on DD.

Finally, we demonstrate a higher prevalence of DD in patients taking metformin compared to those using insulin. While insulin is often associated with more severe disease and higher HbA1C levels, there may be multiple explanations for this finding. First, nearly all patients diagnosed with T1DM are prescribed insulin. The lower prevalence of DD in those with T1DM may have an effect on this result. Additionally, the mechanism of action of each of these drugs may influence these results. Currently, there is limited information on how the pharmacology of each of these

drugs affects DD, and further investigation into the relationship between these drugs and disease pathology could prove to be useful.

Although the size and results of this study are impactful, it has several limitations. TriNetX automatically excludes patient records more than 20 years old, which may introduce bias into our study population. Additionally, patients under the age of 18 are included in our analysis. Our study was designed to capture the largest number of patients possible, and we appreciate that DD is very rare in young children; therefore, it is possible that a DD diagnosis in this age group is an EMR coding error. This study is a database study, and individual patients could not be analyzed. Without access to individual patient records, it is not possible to determine whether a patient's DD or DM diagnosis came first. We are also not aware of the geographical locations from which the EMR data is collected due to privacy regulations by TriNetX. Although we collected demographic data for the population studied, this limitation may preclude us from determining other confounding variables associated with the development of DD or DM. In terms of accounting for confounding variables, significant demographic differences exist in several cohorts analyzed. Although we were able to match groups by age and gender to minimize confounding factors, certain demographic information was not controlled for, as this yielded a sample size that was too small for meaningful comparison. These demographics may play a role in the development of DD, which was not illustrated here. Finally, for all analyses, statistical significance is reported as a result of null hypothesis testing. The large sample size included in the analysis leads to extremely high power, making it necessary to evaluate whether statistically significant results are clinically relevant. Past studies, in addition to the senior author's case logs, as discussed above, provide support for these findings having clinical relevance.

Overall, the reported findings may be explained by the increased accumulation of AGEs associated with diabetes and at a higher level in type 2 diabetes. Future studies investigating the relationship between AGE accumulation and the development and progression of DD may elucidate the connection between DD and DM. Additionally, future studies using the TriNetX database can determine what role factors, including medications, associated diseases such as epilepsy and alcohol use disorder, and other confounding factors play in the pathogenesis of DD.

## Data availability

The data that support the findings of this study are available from TriNetX, but restrictions apply to the availability of these data, which were used under license for the current study, and so are not publicly available. Data are however available from the authors upon reasonable request and with permission of TriNetX. The source data used to generate Figs. 1–4 are available in the Supplementary Data File.

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

## Author contributions

L.C.H. and D.E.K. conceived the project, interpreted data, and revised the manuscript. S.G. performed data collection and analysis. R.P.T. and S.G. drafted the manuscript.

## Competing interests

The authors declare no competing interests.
