## [Peer Review File · Communications Medicine]

Reviewers' comments:

Reviewer #1 (Remarks to the Author):

In this article, the authors aimed to examine the association between DD and DM. This is a very interesting study given its size and the use of registry data, but there are some points that require attention:

Major points that can be fixed:

- I think this paper would benefit from using the RECORD guideline (see EQUATOR network website), so that the context would become clearer.
- Methods, "This was a ...contains de-identified data" I think we would need some more info on this data source. E.g. Which type of health care organisation provide data, how is that being done, which country/countries, etc. This is important to be able to get an idea of generalisability of the results.
- Methods, "Patien records more than....the TriNetX platform": What influence does this may have on the results? I think this should be addressed in the discussion.
- Results, overall: A lot of information is being duplicated in the text and tables. I think the manuscript will improve and also will be more condensed when the authors choose for either presenting it in the text, or in the tables. This is the case for line "There were 100,218... in the T2DM cohort" and Table 1, and all supplementary tables. In my view, the supplementary tables can be removed as they provide no new information.
- I wonder why the authors decided to test all difference using null-hypothesis testing. One may debate on whether it is helpful to use p-values with these large numbers. Because of the large sample, the power is extremely high, resulting easily in statistically significant differences that may not be clinically relevant
- Graph 1: It may be helpful to add confidence intervals to the prevalence rates. I would also suggest to remove the observations below a certain age because it may also be coding errors (DD is very rare in kids). Also: Prevalence is usually expressed as percentage. I assume this graph is showing the proportion of DD?
- To enhance comparison with other studies, it is useful to standardise the prevalence. I think that was not done in this study
- Results, "The majority of DD cases....than the T1DM cohort " I don't understand this line of reasoning. If DM is expected to be associated with DD, it may be expected that DD is more common in T1DM patients (since they were exposed longer to harmful effects of DM). This is not what the observers found. But why is this related to the cohort size? We are comparing prevalences here, which are relative measures, so how can the size of the subcohorts be the reason for this unexpected finding?
- Discussion: "Within the overall...in previous studies": The prevalence of DD diagnoses is also depending on which health care organisations are registering in TriNexT. So more information regarding this is required.
- Discussion: Did the authors also think about the possible effect of medications in the association between DD and DM? This might also explain some of the observations.

Major points that cannot be fixed:

- In the introduction, the authors state that they are interested in the association between DD and DM. I am therefore somewhat surprised that they did not use regression analyses, which would have been logical to use. Besides, the analysis can also be adjusted for potential confounders, which is not the case in the current analyses. Especially when comparing the HbA1c subgroups and T1 and T2 subgroups, it would have be very useful to support the findings with regression analyses. This also avoids losing patients because of matching.

Minor points:

- Introduction, line 1: Reference 1 is not the correct reference to be placed here.
- Introcution, "Moreover, of these studies....systematic review": As one of the authors of this article, I would be very interested in more detailed information what caused the duplication. We did look at data-overlap in this systematic review and excluded all overlapping data that we could identify.
- Results, "When comparing the DM and T2DM cohorts..... than T1DM patients" I find it somewhat

peculiar that the authors decided to compare the T1DM and T2DM cohorts with the overall DM cohort, since these contain partly the same patients. Especially for the T2DM cohort vs. Overall DM cohort, 92% of the overall cohort consists of T2DM patients. It is typical to compare these patients with mostly themselves.

- Results: there are 2 formulas reported in the results, which should be placed in the Methods.
- Discussion: "Previous studies have.... From 1-8.2%": This range is not consistent with the results of the studies that are cited here
- Discussion "As stated previously...from 14-31%": Please note that these studies are not reporting incidence.

Reviewer #2 (Remarks to the Author):

This is a unique study (in numbers) on the relationship between Dupuytren's Disease and Diabetes. I'm sure it's a very interesting topic for the Dupuytren's disease scientific community.

Reviewer #3 (Remarks to the Author):

The authors should be commended on leveraging rich EMR database for examining the relation between Dupuytren's disease (DD) and diabetes mellitus (DM). The manuscript addresses relevant question on developing an understanding around DD's burden overall and in those adults with DM. However, there are some weaknesses stated below that the authors might want to consider to strengthen the paper further:

1. Given the focus of the paper was to understand DD's burden, it unclear, why authors chose to focus on DM only and not explore the role of other risk factors such as smoking, alcoholism etc.
2. The database that authors have leverage seemed powerful since it has data from 65 healthcare organization. The authors may want to comment on how the EMR data was standardized across different providers. For instance, some healthcare providers may use ICD codes vs. other may use SNOMED codes? How did authors account for the heterogeneity in EMR across different healthcare providers? For instance, was EMR data standardized to some form of common data model like OMOP?
3. More details on how authors defined DD and DM using EMR data is needed. Curious to hear authors rationale to simply use condition codes to define the diseases. The authors may want to consider well-established phenotypic definitions to define the diseases. For instance, leverage condition, medication and lab values to define a disease. There are various algorithms in place to define different disease. For instance, <https://phekb.org/phenotype/type-2-diabetes-mellitus> provides Type 2 DM algorithm that can be extracted from EMRs.
4. It is unclear how did authors account for longitudinality. To clarify, it seems authors were interested in computing risk ratio, which refers to risk of developing DD. However, authors show the prevalent cases for DD. Therefore, it unclear where DD came before or after DM. the timing of the both diseases are unclear. Since authors have the longitudinal EMR data, which provides course of healthcare for given person, it might provide an opportunity to explore this further.
5. The authors only report age, sex and race in demographic characteristics. Based on the demographic characteristics, it seems there were race differences too. However, while conducting the comparison, authors only did age and sex match analysis. They may want to include race too in the matching when they explore the association between groups.
6. The results, conclusion and implication are bit confusing. For instance, authors state multiple risk factors but have focused the investigation on DM as a risk factor. It is unclear how other factors may interplay or confound this relationship? More discussion around mechanism for medication is needed – for e.g. why only insulin and metformin were studied? Especially, given

metformins are usually not recommended for those adults who have decreased heart or kidney function or dependence on alcohol.

Minor comments:

1. Recommend authors to refrain from using language such as diabetic patients as it kind of dehumanizes the patients reducing them to their condition. Would recommend authors to reword as patients with diabetes ...as opposed to diabetic patients.

Reviewer #1 (Remarks to the Author):

In this article, the authors aimed to examine the association between DD and DM. This is a very interesting study given its size and the use of registry data, but there are some points that require attention:

Major points that can be fixed:

- I think this paper would benefit from using the RECORD guideline (see EQUATOR network website), so that the context would become clearer.

We have reviewed the RECORD guidelines and the only component we did not include was the type of access we had to the patient data within TriNetX. This point has now been addressed in the methods section on lines 48-49.

- Methods, “This was a ...contains de-identified data” I think we would need some more info on this data source. E.g. Which type of health care organisation provide data, how is that being done, which country/countries, etc. This is important to be able to get an idea of generalisability of the results.

We utilized TriNetX’s research network, which includes data from 76 healthcare organizations across 4 countries. We do not have access to the locations of the healthcare organizations due to TriNetX privacy regulations.

In terms of how data is collected and included on the TriNetX database, TriNetX offers the following explanation: “TriNetX typically receives data from HCOs and other data providers in one of two ways: TriNetX ingests data directly from an HCO’s research repository (e.g., i2b2) into the TriNetX environment. An HCO sends TriNetX data extracts in the form of CSV files. TriNetX maps the data to a standard and controlled set of clinical terminologies. The data is then transformed into a proprietary data schema. This transformation process includes an extensive data quality assessment that includes ‘data cleaning’ that rejects records that don’t meet the TriNetX quality standards.”

Further information regarding TriNetX’s data can be found at the following website (which has also been added to the methods section on lines 45-46):

<https://support.trinetx.com/hc/en-us/sections/360000928753-About-the-Data>

- Methods, “Patient records more than 20 years old were automatically excluded from analysis from the TriNetX platform”: What influence does this may have on the results? I think this should be addressed in the discussion.

We agree that this may have an influence on the results and have addressed it in the discussion on lines 283-284.

- Results, overall: A lot of information is being duplicated in the text and tables. I think the manuscript will improve and also will be more condensed when the authors choose for either presenting it in the text, or in the tables. This is the case for line “There were 100,218... in the T2DM cohort” and Table 1, and all supplementary tables. In my view, the supplementary tables can be removed as they provide no new information.

This line has been removed from the results section, and supplementary tables have also been removed, with the exception of Supplementary Table 6, which has been changed to Supplementary Table 1. This table provides yearly prevalence data not presented anywhere else in the manuscript.

- I wonder why the authors decided to test all difference using null-hypothesis testing. One may debate on whether it is helpful to use p-values with these large numbers. Because of the large sample, the power is extremely high, resulting easily in statistically significant differences that may not be clinically relevant

We agree that the p-values are low with large sample sizes, but null hypothesis testing is standard for the statistical tests used in our analysis. We have added a note to address the concern of statistical significance versus clinical significance in the discussion on lines 297-301.

- Graph 1: It may be helpful to add confidence intervals to the prevalence rates. I would also suggest to remove the observations below a certain age because it may also be coding errors (DD is very rare in kids). Also: Prevalence is usually expressed as percentage. I assume this graph is showing the proportion of DD?

The prevalence rates were generated by the analytical tools on TriNetX, which do not allow for the calculation of confidence intervals. The graph previously showed the proportion of DD; however, to standardize the prevalence, we have changed all of our reported values to percentages. These changes can be seen in Figures 2 and 3. Furthermore, we have added a note in the discussion to address the inclusion of patients under the age of 18 on lines 284-287.

- To enhance comparison with other studies, it is useful to standardise the prevalence. I think that was not done in this study

Thank you for this suggestion. We are changing our reported values to percentages as can be seen in Figures 2 and 3.

- Results, “The majority of DD cases....than the T1DM cohort “ I don't understand this line of reasoning. If DM is expected to be associated with DD, it may be expected that DD is more

common in T1DM patients (since they were exposed longer to harmful effects of DM). This is not what the observers found. But why is this related to the cohort size? We are comparing prevalences here, which are relative measures, so how can the size of the subcohorts be the reason for this unexpected finding?

This line in the results is referring to the absolute number of DD cases as opposed to prevalence of DD. As shown in Supplemental Table 1, there are more cases of DD within the population of patients with T2DM than in the population of patients with T1DM. This higher absolute number of DD cases is related to the size of the subcohorts, with the T2DM subcohort being much larger than the T1DM cohort.

- Discussion: “Within the overall...in previous studies”: The prevalence of DD diagnoses is also depending on which health care organisations are registering in TriNexT. So more information regarding this is required.

We do not have access to the locations of the healthcare organizations due to TriNetX privacy regulations. We have added a note in the discussion to address this point on lines 289-293.

- Discussion: Did the authors also think about the possible effect of medications in the association between DD and DM? This might also explain some of the observations.

Yes, we are interested in looking at the effect of medications other than insulin and metformin in addition to how their mechanisms of action may be related to the development of DD in patients with diabetes. We have added a sentence in the conclusion indicating this as an anticipated direction of future research (lines 316-319). We are saving a more detailed analysis for future work as the intention of this paper was to focus on general trends.

Major points that cannot be fixed:

- In the introduction, the authors state that they are interested in the association between DD and DM. I am therefore somewhat surprised that they did not use regression analyses, which would have been logical to use. Besides, the analysis can also be adjusted for potential confounders, which is not the case in the current analyses. Especially when comparing the HbA1c subgroups and T1 and T2 subgroups, it would have been very useful to support the findings with regression analyses. This also avoids losing patients because of matching.

Thank you, we will consider this moving forward.

Minor points:

- Introduction, line 1: Reference 1 is not the correct reference to be placed here.

Thank you, we have updated our references with the correct citation.

- Introduction, “Moreover, of these studies....systematic review”: As one of the authors of this article, I would be very interested in more detailed information what caused the duplication. We did look at data-overlap in this systematic review and excluded all overlapping data that we could identify.

The individual systematic review did not duplicate patients; however, in our literature search, patients we had previously described in other studies were also included in the review. We have clarified what is meant by duplication on lines 29-31 of the introduction.

- Results, “When comparing the DM and T2DM cohorts..... than T1DM patients” I find it somewhat peculiar that the authors decided to compare the T1DM and T2DM cohorts with the overall DM cohort, since these contain partly the same patients. Especially for the T2DM cohort vs. Overall DM cohort, 92% of the overall cohort consists of T2DM patients. It is typical to compare these patients with mostly themselves.

We agree with this comment and have removed the demographic comparisons for T1DM vs. DM and T2DM vs. DM.

- Results: there are 2 formulas reported in the results, which should be place in the Methods.

We have moved these formulas from the results to the methods section (moved to lines 70-71).

- Discussion: “Previous studies have.... From 1-8.2%”: This range is not consistent with the results of the studies that are cited here

We have double-checked these studies and corrected the prevalence range on line 198.

- Discussion “As stated previously...from 14-31%”: Please note that these studies are not reporting incidence.

We have double-checked these studies and corrected line 241 as the studies report prevalence, not incidence.

Reviewer #2 (Remarks to the Author):

This is a unique study (in numbers) on the relationship between Dupuytren's Disease and Diabetes. I'm sure it's a very interesting topic for the Dupuytren's disease scientific community.

Thank you!

Reviewer #3 (Remarks to the Author):

The authors should be commended on leveraging rich EMR database for examining the relation between Dupuytren's disease (DD) and diabetes mellitus (DM). The manuscript addresses relevant question on developing an understanding around DD's burden overall and in those adults with DM. However, there are some weaknesses stated below that the authors might want to consider to strengthen the paper further:

1. Given the focus of the paper was to understand DD's burden, it unclear, why authors chose to focus on DM only and not explore the role of other risk factors such as smoking, alcoholism etc.

The focus of this paper was to specifically look at the relationship between DD and DM. Analysis of other risk factors is outside of the scope of the current study; however, we are intending to complete similar analyses in the future on the relationship between DD and these risk factors. This has been noted as a future direction of research in the conclusion, lines 316-319.

2. The database that authors have leverage seemed powerful since it has data from 65 healthcare organization. The authors may want to comment on how the EMR data was standardized across different providers. For instance, some healthcare providers may use ICD codes vs. other may use SNOMED codes? How did authors account for the heterogeneity in EMR across different healthcare providers? For instance, was EMR data standardized to some form of common data model like OMOP?

The EMR data obtained from the healthcare organizations are standardized by the TriNetX network prior to publication. TriNetX offers the following explanation: "TriNetX typically receives data from HCOs and other data providers in one of two ways: TriNetX ingests data directly from an HCO's research repository (e.g., i2b2) into the TriNetX environment. An HCO sends TriNetX data extracts in the form of CSV files. TriNetX maps the data to a standard and controlled set of clinical terminologies. The data is then transformed into a proprietary data schema. This transformation process includes an extensive data quality assessment that includes 'data cleaning' that rejects records that don't meet the TriNetX quality standards."

3. More details on how authors defined DD and DM using EMR data is needed. Curious to hear authors rationale to simply use condition codes to define the diseases. The authors may want to consider well-established phenotypic definitions to define the diseases. For instance, leverage condition, medication and lab values to define a disease. There are various algorithms in place to

define different disease. For instance, <https://phekb.org/phenotype/type-2-diabetes-mellitus> provides Type 2 DM algorithm that can be extracted from EMRs.

The use of the suggested algorithm to define DM would require access to individual patient EMR data, which unfortunately is not available on the TriNetX platform. TriNetX presents aggregate EMR data to maintain patient privacy. Additionally, there is no definitive diagnostic test or lab value that confirms DD, therefore the CPT codes were the only way to identify these cases using this platform.

4. It is unclear how did authors account for longitudinality. To clarify, it seems authors were interested in computing risk ratio, which refers to risk of developing DD. However, authors show the prevalent cases for DD. Therefore, it is unclear where DD came before or after DM. The timing of the both diseases is unclear. Since authors have the longitudinal EMR data, which provides course of healthcare for given person, it might provide an opportunity to explore this further.

TriNetX only presents aggregate EMR data, which does not allow for the relative timing/onset of DD and DM to be specified for each patient. This is a limitation of our analysis that we have now addressed on lines 288-289 in the discussion.

5. The authors only report age, sex and race in demographic characteristics. Based on the demographic characteristics, it seems there were race differences too. However, while conducting the comparison, authors only did age and sex match analysis. They may want to include race too in the matching when they explore the association between groups.

When matching for age, sex, and race in the risk ratio calculations, the sample size of the cohorts being studied became too small for meaningful comparison. This is a limitation of our analysis that we have now addressed on lines 293-296 in the discussion.

6. The results, conclusion and implication are bit confusing. For instance, authors state multiple risk factors but have focused the investigation on DM as a risk factor. It is unclear how other factors may interplay or confound this relationship? More discussion around mechanism for medication is needed – for e.g. why only insulin and metformin were studied? Especially, given metformins are usually not recommended for those adults who have decreased heart or kidney function or dependence on alcohol.

There are multiple risk factors that play a role in the development of DD; however, we are saving a more detailed analysis for future work as this paper was more focused on general trends. We have added a sentence in the conclusion indicating this as an anticipated direction of future research (lines 316-319).

Minor comments:

1. Recommend authors to refrain from using language such as diabetic patients as it kind of dehumanizes the patients reducing them to their condition. Would recommend authors to reword as patients with diabetes ...as opposed to diabetic patients.

Thank you for this recommendation! We have revised all instances of the term “diabetic patients” to “patients with diabetes” throughout the paper to use less stigmatizing language.

Reviewers' comments:

Reviewer #1 (Remarks to the Author):

I am satisfied with the changes that the authors made regarding the reviewer comments.

Reviewer #3 (Remarks to the Author):

I appreciate authors addressing most of the comments raised. I have one additional comment that needs attention:

1. Authors now mentioned explicitly their inability to estimate the development of disease. With this limitation, it is unclear how authors can compute risk since risk is commonly used for incident diseases. Are authors in this study referring to prevalent risk ratio? This is extremely important for readers to not confuse with the cause-effect relationship but rather interpret the findings as association.

Reviewer #1 (Remarks to the Author):

I am satisfied with the changes that the authors made regarding the reviewer comments.

We are glad you are satisfied with our revisions.

Reviewer #3 (Remarks to the Author):

I appreciate authors addressing most of the comments raised. I have one additional comment that needs attention:

1. Authors now mentioned explicitly their inability to estimate the development of disease. With this limitation, it is unclear how authors can compute risk since risk is commonly used for incident diseases. Are authors in this study referring to prevalent risk ratio? This is extremely important for readers to not confuse with the cause-effect relationship but rather interpret the findings as association.

We agree that this study was not designed to assess risk or any cause-effect relationship, and the phrasing of our findings in terms of risk and risk ratio was improper. It is correct that we are looking at prevalence and prevalence ratios rather than risk. We have revised the language throughout the manuscript to reflect this and appreciate the constructive comments.

REVIEWERS' COMMENTS:

Reviewer #3 (Remarks to the Author):

I appreciate that the authors have succinctly addressed the comments/concerns.